# Privacy nudges for disclosure of personal information: A systematic literature review and meta-analysis

**Athina Ioannou**[1]*, **Iis Tussyadiah**[2], **Graham Miller**[2], **Shujun Li**[3], **Mario Weick**[4]

**1** Surrey Business School, University of Surrey, Guildford, United Kingdom, **2** School of Hospitality and Tourism Management, University of Surrey, Guildford, United Kingdom, **3** School of Computing, University of Kent, Canterbury, United Kingdom, **4** Department of Psychology, Durham University, Durham, United Kingdom

* a.ioannou@surrey.ac.uk

## Abstract

### Objective

Digital nudging has been mooted as a tool to alter user privacy behavior. However, empirical studies on digital nudging have yielded divergent results: while some studies found nudging to be highly effective, other studies found no such effects. Furthermore, previous studies employed a wide range of digital nudges, making it difficult to discern the effectiveness of digital nudging. To address these issues, we performed a systematic review of empirical studies on digital nudging and information disclosure as a specific privacy behavior.

### Method

The search was conducted in five digital libraries and databases: Scopus, Google Scholar, ACM Digital Library, Web of Science, and Science Direct for peer-reviewed papers published in English after 2006, examining the effects of various nudging strategies on disclosure of personal information online.

### Results

The review unveiled 78 papers that employed four categories of nudge interventions: *presentation*, *information*, *defaults*, and *incentives*, either individually or in combination. A meta-analysis on a subset of papers with available data ($n = 54$) revealed a significant small-to-medium sized effect of the nudge interventions on disclosure (Hedges' $g = 0.32$). There was significant variation in the effectiveness of nudging ($I^2 = 89\%$), which was partially accounted for by interventions to increase disclosure being more effective than interventions to reduce disclosure. No evidence was found for differences in the effectiveness of nudging with *presentation*, *information*, *defaults*, and *incentives* interventions.

### Conclusion

Identifying ways to nudge users into making more informed and desirable privacy decisions is of significant practical and policy value. There is a growing interest in digital privacy

**Data Availability Statement:** All relevant data are within the manuscript and its Supporting Information files.

**Funding:** This work was part of the "PRIvacy-aware Personal Data Management and Value

Enhancement for Leisure Travellers" (PriVELT) Project, supported by the UK's Engineering and Physical Sciences Research Council (EPSRC), in collaboration with University of Kent, University of Warwick and University of Durham (Grants: IT: EP/R033196/1, SL: EP/R033749/1, and EP/R033609/1). https://epsrc.ukri.org/ The funders had no role in study design, data collection and analysis, decision to publish, or preparation of the manuscript.

**Competing interests:** The authors have declared that no competing interests exist.

nudges for disclosure of personal information, with most empirical papers focusing on nudging with presentation. Further research is needed to elucidate the relative effectiveness of different intervention strategies and how nudges can confound one another.

## 1. Introduction

Human decision making is inherently complex. Immersed in digital environments and while performing online activities, individuals are faced daily with numerous privacy and security decisions: configuring visibility in social networking sites, allowing access to sensitive data in mobile apps, clicking or ignoring links embedded in emails, and so forth. Various factors such as heuristics, cognitive and behavioral biases, and incomplete and asymmetric information can affect privacy decisions and behaviors, often leading to deficient and regrettable choices, such as oversharing, that are not aligned with users' own intentions [1]. Habitual responses and low awareness of ransomware and scams, for example, can result in users disclosing sensitive personal information to unauthorized sources, such as employees giving untrusted third-party sources access to confidential corporate data [2]. These privacy failures impact not only the individuals and organizations directly involved in the collection and use of personal information, but also the economy and society at large due to business disruption, loss of productivity, and other financial and social ramifications. Therefore, finding ways to encourage users to make more informed privacy decisions is of significant practical and policy importance.

Studies have suggested the use of paternalistic interventions or nudges to guide and assist people into changing their security and privacy behaviors in digital environments [1]. A nudge is "any aspect of the choice architecture that alters people's behavior in a predictable way without forbidding any options or significantly changing their economic incentives" [3](p.6). In other words, nudging can influence decision making allowing people to take any course of action, "without limiting individual choice, but in fact preserving freedom of choice" [1](p.44:11). Existing research on nudging has been conducted mostly in offline environments [4]. However, there is a growing body of literature on the use of digital nudging in the context of individuals' privacy-related behavior and decisions. Digital nudging refers to the use of interface elements aiming to guide users to make more informed judgements and decisions in digital choice environments [4, 5].

Empirical studies on the effectiveness of digital nudging on user behaviors are not fully conclusive. Whilst some studies have shown that digital nudging can change user behaviors [6] some others have found no such effect [7]. This may not be a complete surprise as there are vast discrepancies in the type of nudges that have been studied. In addition, outcome behaviors differ widely. For example, Halevi, Kuppusamy, Caiazzo, and Memon [8] sought to nudge users to increase information disclosure, while Bravo-Lillo et al. [9] sought to nudge users to reduce the likelihood of installing malicious software.

As this brief discussion highlights, there is an urgent need for a systematic review and meta-analysis of the literature to establish the effectiveness of nudging interventions in changing privacy-related behaviors and decisions (cf. [10]). In addition, the varied nature of nudging interventions calls for a framework to classify nudging strategies. Acquisti et al. [1] provided such a conceptual framework, with a focus on privacy nudging, which is particularly pertinent for the present investigation. They argued for a conceptual distinction between digital nudges relying on information, defaults, incentives, timing, reversibility or presentation techniques, either individually or in combination. These dimensions were suggested to address various pitfalls when users make privacy decisions, including issues associated with cognitive biases and

information asymmetry. In the present study, we adopt Acquisti et al.'s [1] framework to establish the effects of digital nudges on privacy decisions in digital environments. We focus on disclosure of personal information as a specific type of privacy decisions. Information disclosure (also known as data disclosure or online disclosure) refers to the act of making personal information accessible to other interested parties. We define personal information as any information that can identify an individual directly or indirectly, such as biographical information, telephone number and email address, workplace data and education information, location data, physical, physiological, genetic, mental, economic, cultural or social data of a person [11]. Through our focus on personal information disclosure, we extend a recent meta-analysis on the effectiveness of nudging in online and offline environments [12]. The present work casts a much wider net than Hummel and Maedche's [12], which only examined studies citing [3], thus providing a more comprehensive assessment of the literature.

## 2. Method

This systematic literature review was conducted following the guidelines proposed by PRISMA (2015) as well as by Kitchenham and Charters [13] who have adapted the PRISMA guidelines in the Software Engineering context. The most critical element of this study, the review protocol, was developed first. The protocol was published in a networking portal for scientists, ResearchGate (https://www.researchgate.net/). The study followed the steps outlined in the review protocol, which are discussed in the following subsections.

### 2.1. Search strategy

The research question addressed in this systematic review is: "What are the effects of various intervention (nudging) strategies on disclosure of personal information online?" The search was conducted in five digital libraries and databases: Scopus, Google Scholar, ACM Digital Library, Web of Science, and Science Direct. All searches were based on title, abstract, and keywords, and took place in September 2020. The search strategy and the combination of keywords used in this study is presented in Table 1. Please note that prod refers to experimental design. Prod is used as a synonym to nudge.

**Table 1. Search strategy.**

| Search Strategy | #1 AND #2 AND #3 AND #4 AND #5 |
|---|---|
| Concepts | Keywords |
| #1 Privacy | "privacy" OR "confidential*" OR "security" |
| #2 Personal Information | "personal information" OR "personal data" OR "sensitive information" OR |
| | "sensitive data" OR personal information OR personal data OR sensitive |
| | information OR "private information" OR "private data" |
| #3 Information Disclosure | "information disclosure" OR "willingness to disclose" OR "intention to |
| | disclose" OR "likelihood to disclose" OR "willingness to share" OR "intention |
| | to share" OR "data sharing" OR "likelihood to share" OR "disclosure |
| | behavio*" OR "data disclosure" OR "online disclosure" OR shar* OR "self- |
| | disclosure" OR "online information sharing" |
| #4 Nudging | "nudge*" OR "nudging" OR "intervention*" OR "experiment*" OR |
| | "paternalis*" OR "prod" OR "randomi* control trial" OR "quasi-experiment" |
| | OR |
| | "choice architecture" OR "default" OR "framing" OR "priming" OR |
| | "incentive*" OR "monet*" |
| #5 Online | "online" OR "internet" OR "web" OR "digital" OR "software" |

The search strategy depicted in Table 1 was used as a template for the searches in all five databases. As each one of the digital libraries has a different search engine, a preliminary search was conducted on each digital library adapting the search terms to the requirements of the relevant search engine. More details on the exact search terms used in each database can be found in S1 Appendix. Since in the last decade the privacy of individuals has been affected by new technological solutions such as smartphones and Internet of things, papers published before 2006 are excluded because the meaning of privacy might have changed in the years following 2006 [14]. All details of papers that were considered potentially relevant were managed using the reference management tool Mendeley (version 1.19).

## 2.2. Study selection

Two co-authors (reviewers, hereafter) conducted searches in all five digital libraries. Each reviewer performed an initial screening of their search results based on titles and abstracts. The results of the initial screening were compared between reviewers to check for consistency and discrepancies in the search process.

**2.2.1. Inclusion criteria.** Studies evaluating the effects of various intervention (nudging) strategies on the disclosure of personal information in the context of online privacy were of interest in the present review. Thus, the following inclusion criteria were applied:

- Empirical studies are eligible for inclusion. Empirical studies reporting experimental manipulations and quasi-experimental variations are considered eligible, as well as studies conducted in the laboratory, field, and online.

- Studies that include technology usage, such as internet, social media, e-commerce websites, mobile phones, and other digital platforms, are considered eligible. There are no age restrictions for participants. Where information is available on individuals' health status, studies of interest include healthy participants.

- Studies using one or more of the following intervention strategies (nudges) as mentioned in the work of Acquisti et al. [1] are eligible: information (feedback and education), presentation (framing, ordering, salience and structure), default, timing, reversibility and incentive (priming).

- Studies that depict the intervention strategies as independent variable(s) and intention/willingness to disclose or share personal information as well as actual disclosure or sharing behavior as dependent variable(s) are of interest. Studies that include antecedents of dependent and independent variable(s) as well as mediators and/or moderators in this relationship are also eligible.

**2.2.2. Exclusion criteria.** Papers that meet the following criteria were excluded from the review:

- Papers published in languages other than English.

- Theoretical or conceptual publications related to intervention strategies and individual privacy.

- Papers reporting studies conducted in clinical settings or with clinical samples (e.g., visually impaired individuals) or special populations (e.g., abuse survivors).

- Papers reporting studies investigating the effect of nudging interventions on outcomes other than information disclosure, such as password creation, selection of secure Wi-Fi, intention towards policy compliance, intention to install software applications, and similar others.

In the event the primary reviewer was unsure regarding the application of inclusion and exclusion criteria in a particular study, the opinion of the second primary reviewer was sought in order to reach a decision. Failure to meet any one of the above eligibility criteria resulted in exclusion from the review and any apparent discrepancies during the selection process was resolved through discussion or, in case no agreement was reached, through consultation of an independent reviewer. The number of excluded papers, including reasons of exclusion for those excluded following review of the full text, was recorded at each stage.

### 2.3. Study quality assessment

There have been several quality checklists published in extant academic literature, although most of them addressed medical studies. Aiming to rigorously assess the methodological quality of the studies included in this systematic review, we followed the guidelines of Kitchenham and Charters [13] as well as the systematic review of Zhou *et al.*, [15] on the existing quality assessment tools that are being used in systematic reviews in the area of Software Engineering. Both studies recommended a set of questions, derived from the most commonly used checklists and guidelines regarding the design, conduct, analysis, and conclusions of each study included in this systematic review. According to the literature, researchers should review the available list of questions in the context of their own study and select the most appropriate evaluative questions for their study [13]. For this reason, questions from different checklists that have been used in previous research were reviewed and integrated for the purposes of this systematic review. The criteria to evaluate each study were based on the evaluative questions presented in Table 2.

The scoring procedure was Y = 1 and N = 0. Studies could reach a minimum of 0 and maximum 10 points. We established a cut-off of 6 as it represents 60% of all questions in the quality assessment [16]. Papers receiving a score exceeding 6 (>6) were decided to be retained in this systematic review (see S1 Appendix).

### 2.4. Data extraction

The following data were extracted from each paper: full reference (including name(s) of author(s), year of publication, and publication venue), description of the intervention strategy, dependent variable(s) (e.g., intention to disclose information or actual disclosure behavior), antecedent, moderator, and mediator variables(s), main findings, type of study (e.g., experiment), type of statistical analysis (e.g., regression), type of outcome variable (e.g., dichotomous, continuous), randomness of allocation, and existence of control group. Specifically, for studies

**Table 2. Quality assessment questions.**

| No. | Question |
|---|---|
| **QA1** | Are the aims of the research clearly defined? |
| **QA2** | Is there an adequate description of the context in which the research was carried out? |
| **QA3** | Was the research design appropriate to address the aims of the research? |
| **QA4** | Was there a control group with which to compare treatments? |
| **QA5** | Are the data collection methods adequately described? |
| **QA6** | Were all measures used in the study fully defined? |
| **QA7** | Is the experimental design appropriate and justifiable? |
| **QA8** | Does the study provide description and justification of the data analysis approaches? |
| **QA9** | Are the findings of the study clearly stated? |
| **QA10** | Does the study add value to academia or practice? |

that were considered potential candidates for inclusion in a meta-analysis, we recorded effect sizes or any other data that would allow for the calculation of an effect size.

Each study included in the systematic review was reviewed by two co-authors. One acted as the main data extractor, while the second acted as the data checker. The first reviewer was responsible for extracting the data from all studies that were selected for inclusion, while the second reviewer was responsible for checking if the data extracted by the former were correct. Any apparent disagreements during the data extraction process were resolved through discussion and, in case that no agreement could be reached, the consultation of an independent reviewer was sought.

## 2.5. Statistical analysis

We performed a meta-analysis on papers included in the systematic review, which reported appropriate effect sizes or data that enabled us to calculate effect size statistics. As a result, a further evaluation of the selected studies was performed in order to select studies that fulfil the requirements necessary for a meta-analysis. The selection of papers that would be included in the meta-analysis was based on whether the papers satisfied the following criterion [17]: Papers were considered eligible for meta-analysis only if they reported results including an effect size (Cohen's *d*, Odds Ratio, Pearson's *r*, Chi squared, eta squared) or provided enough information for the computation of one (means and standard deviations, the percentage of participants in treatment and control groups, F values, regression coefficients). When the reported results did not enable the computation of an effect size, the paper was excluded from meta-analysis [18].

## 3. Results and discussion

### 3.1. Search results

Integrating the results derived from all five digital libraries, 2,046 papers were identified before duplicate papers were removed. More specifically, each database yielded the following results: 343 papers from Scopus, 159 papers from Science Direct, 314 papers from ACM Digital Library, 769 papers from Web of Science, and 459 papers from Google Scholar. Moreover, additional papers were identified by scanning reference lists and citations of prominent articles in the investigated subject, a search approach also known as snowballing. It has been suggested that researchers can achieve the best possible coverage of existing literature by using snowballing as a complementary approach to database search [19]. The snowballing process resulted in 106 papers. We screened the title, abstract, and keywords of 2,046 + 106 = 2,152 papers identified through database searches and snowballing. We excluded papers that were deemed irrelevant or duplicates. As a result, a total of 254 papers passed the initial screening and were subject to a more detailed assessment. The full list of papers (*n* = 254) was then evaluated against a specific set of inclusion and exclusion criteria. In case of similar studies using the same dataset, we retained the one providing more detailed information. After evaluating the inclusion and exclusion criteria, 150 papers were excluded, with 104 papers remaining for full text assessment. The full text screening resulted in 39 papers identified for exclusion, mostly due to the fact that the outcome variable in the study was not related to information disclosure or the employed intervention cannot be classified as a nudge. Other reasons for exclusion include papers conducting surveys instead of experiments, papers investigating information disclosure using factors other than nudging/interventions as well as papers that have published the same results in different outputs. As a result, a total of 78 papers were included in the present systematic review (see Fig 1).

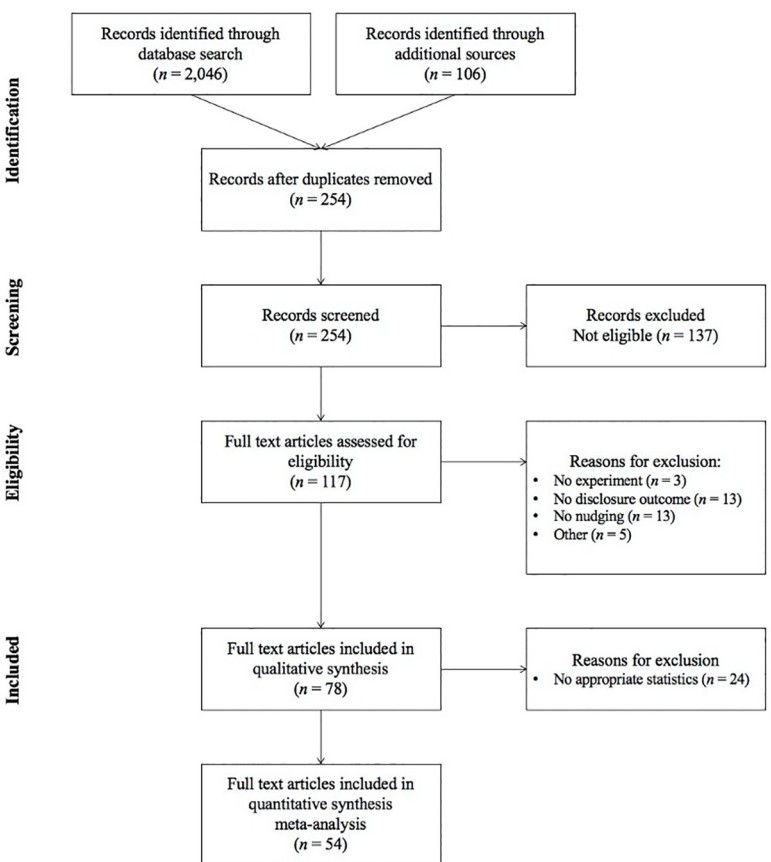

**Fig 1. PRISMA flow diagram of information through different stages of review.**

## 3.2. Study characteristics

Overall, the final selection of papers included in the qualitative review suggests that during the last decade there has been a surge of interest in the effects of nudges on privacy decisions (see Fig 2). More specifically, during the years 2013–2020, there has been a dramatic increase in scientific publications ($n = 64$), while during the years 2006–2012 the published papers in the area were quite scarce ($n = 14$), and in 2008 there were no publications at all. As a result, we can infer that the investigation of privacy nudges has grown significantly during the last decade, most likely since 2011–2012, taking into consideration the time delay between investigations and academic publications.

Regarding the type of publications, most papers were published in academic journals ($n = 43$), while almost half of the included papers were published in conference proceedings ($n = 35$), thus signifying a balance in the publication outputs of the included papers. Furthermore, regarding the type of participants, most papers ($n = 35$) used commercial platforms to recruit participants (e.g., Amazon Mechanical Turk), while several papers drew on student samples ($n = 15$) in their experiments. Regarding the origin of participants, most papers were conducted in the USA ($n = 24$) and in Europe ($n = 12$), although a majority of papers did not reveal the location of the experiment or the origin of the recruited participants ($n = 30$).

Table 3 presents the studies included in the qualitative part of this systematic review ($n = 78$) along with author(s) and publication year, intervention or nudging strategy, and main findings of the study. More detailed information regarding the papers can be found in

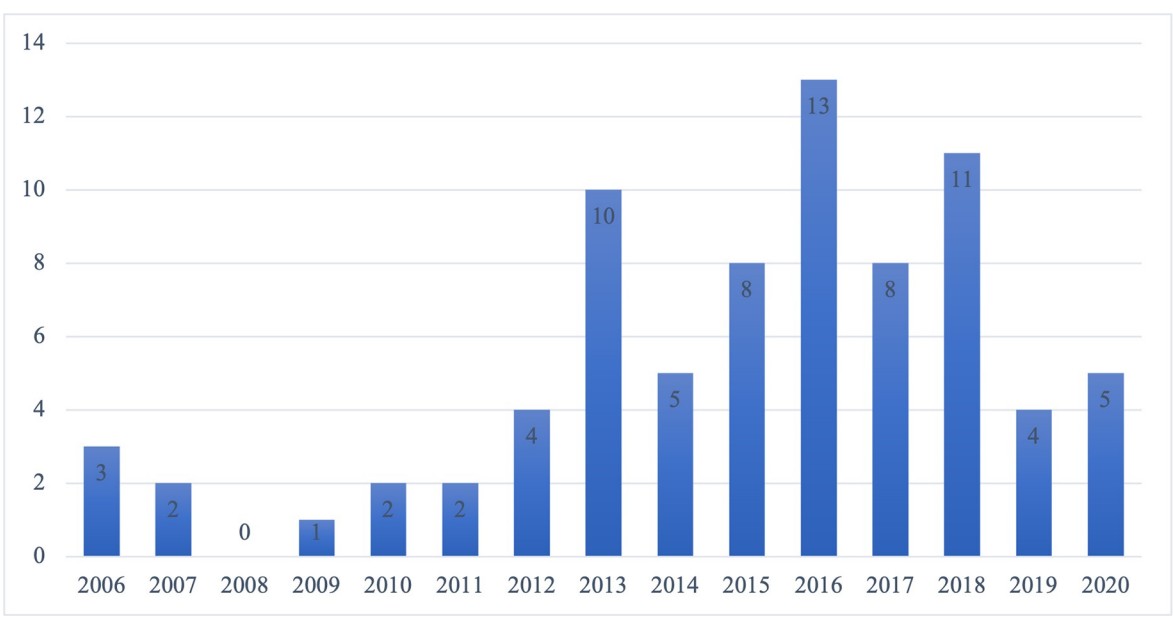

**Fig 2. Number of publications per year.**

S1 Appendix. Overall, it can be inferred that the majority of papers employed factorial between-subjects experimental designs as well as surveys that include scenarios in order to test the effects of nudging on intended or actual information disclosure of individuals.

The present systematic review focuses on the effects of intervention strategies on information disclosure of individuals in online contexts. Thus, in order to explicate the effects of different nudging strategies, we grouped studies into four nudging categories according to the dimensions put forward by Acquisti et al. (2017) [1]: information, presentation, defaults, and incentives. Reversibility and timing, which are part of the nudging dimensions suggested by Acquisti et al. (2017) [1], were initially taken into consideration. However, during the search process we identified no papers under the timing dimension, and only one paper that could be classified into reversibility as well as presentation nudging. According to Acquisti et al. (2017), the dimensions of the framework are not mutually exclusive. For purposes of comparability, we classified the paper under presentation nudging, thus covering the four dimensions in the framework. As some papers have used a combination of the aforementioned four categories of nudges, a fifth category (i.e., combination) was added. As seen in Table 3, it is apparent that most papers (47 out of 78 papers) used presentation interventions to influence privacy decisions and disclosure of personal information, followed by combinations of the four interventions (fourteen papers), information (nine papers), incentives (four papers) and defaults (four papers).

## 3.3. Qualitative synthesis

**3.3.1. Nudging with presentation.** Nudging with presentation refers to how information and choices are presented to users using design and delivery concepts such as framing, ordering of alternatives, saliency, and structure of presented information. Examples include reminders, privacy notices and warnings, design features, and visual cues. Forty-one papers were identified employing presentation as a nudging intervention aiming to explore the influence on information disclosure, however, the majority of extant literature has offered mixed and contradictory findings.

**Table 3.  Studies categorized by intervention (nudging) strategy.**

| Author | Intervention | Main Findings |
|---|---|---|
| *Presentation* | | |
| Acquisti, A., John, L. K., & Loewenstein, G. (2012) | Feedback on other's admissions, presentation—intrusiveness order | Self-disclosure is affected by information about others' divulgences and the mere order in which sensitive inquiries are presented. |
| Adjerid, I., Acquisti, A., Brandimarte, L., & Loewenstein, G (2013) | Reference dependence and framing, and salience of privacy notices | The impact of privacy notices on disclosure is sensitive to saliency and framing while misdirections reduce the impact of privacy notices on disclosure. |
| Adjerid, I., Peer, E., & Acquisti, A (2018) | Normative and behavioral factors: objective and relative changes in (levels) of privacy protection | Both objective and relative changes in privacy protection can influence participants' self-disclosure behavior. |
| Alemany, J., del Val, E., Alberola, J. and García-Fornes, A. (2019) | Picture nudge, number nudge | Nudges can increase user privacy awareness in social networks. |
| Babula, E., Mrzygłód, U., & Poszewiecki, A (2017) | Priming | Priming has a negative impact on willingness to share one's data. |
| Balebako, R., Péer, E., Brandimarte, L., Cranor, L., & Acquisti, A (2013) | Cognitive disfluency | Disfluency does not reliably or consistently affect self-disclosure. |
| Becker, M., Matt, C. and Hess, T. (2020) ' | Persuasive message: attribute framing, argument strength | Individuals who receive messages with positive framing and argument strength are more likely to disclose personal health information. |
| Ben-shahar, O., & Chilton, A (2016) | Simplification of privacy disclosures | Simplification techniques have little effect on respondents' comprehension of the disclosure, and willingness to share personal information. |
| Betzing, J. H., Tietz, M., vom Brocke, J. and Becker, J. (2020) | Transparency in permission requests | Increased transparency does not decrease the number of users who consent to data processing |
| Bhatia, J., Breaux, T. D., Reidenberg, J. R., & Norton, T. B (2016) | Vagueness in statement, risk likelihood | Findings show how increases in vagueness decrease users' acceptance of privacy risk and thus decrease users' willingness to share personal information. |
| Brandimarte, L., Acquisti, A., & Loewenstein, G (2013) | Control (release of information and access to information) | Perception of control affected individuals' privacy concern to the point that their willingness to disclose sensitive information increased. |
| Carpenter, S., Shreeves, M., Brown, P., Zhu, F., & Zeng, M (2018) | Warnings | Warnings can reduce disclosure of personal information. |
| Carpenter, S., Zhu, F., & Kolimi, S (2014) | Warnings | Warnings are effective in decreasing information disclosure. |
| Carpenter, S., Zhu, F., Zeng, M., & Shreeves, M (2017) | Warnings with sources | Warnings with sources can reduce the extent of disclosure. |
| Chang, D., Krupka, E. L., Adar, E., & Acquisti, A (2016) | Norm shaping design patterns | Design patterns shape perceptions that lead to change of behavior (sharing information). |
| Eling, N., Rasthofer, S., Kolhagen, M., Bodden, E., & Buxmann, P(2016) | Coarse- and fine-grained requests | By displaying fine-grained information, users' likelihood to disclose their information seems to be substantially lowered. |
| Gerlach, J., Widjaja, T., & Buxmann, P. (2015) | Permissiveness of privacy policies | A privacy policy's permissiveness is negatively related to users' willingness to disclose personal information. |
| Hanson, J., Wei, M., Veys, S., Kugler, M., Strahilevitz, L. and Ur, B. (2020). | Hyper personalised ad in robotext or banner | People reacted negatively in the hyper personalised advertisement. However, people continued disclosing their personal information although feeling angered or shocked by the ad. |
| Hughes-Roberts, T. (2015) | Privacy salient information | One form of salience can be particularly effective in persuading users at the point of interaction through dynamic UI elements that instantly. |
| Ilany Tzur, N., Zalmanson, L., & Oestreicher-Singer, G (2016) | Calls to action | Exposure to calls to action can increase the propensity to reveal personal information |
| John, L. K., Acquisti, A., & Loewenstein, G. (2011) | Contextual cues | Contextual cues increase disclosure of personal information. |
| Keith, M. J., Fredericksen, J. T., Reeves, K. S., & Babb, J (2018) | Video privacy policies | The most effective privacy policy videos are those using female narrators with vibrant color palettes and light musical tones. |
| Kim, J., Gambino, A., Sundar, S., Rosson, M., Aritajati, C., Ge, J., & Fanning, C. (2018) | Visual cues, community frame | Interface cues implying greater crowd size and connectivity lead to more self-disclosure of sensitive information, while the community frame has no effect on self-disclosure. |
| Knijnenburg, B. P., Kobsa, A., & Jin, H. (2013) | Fine grained and coarse-grained options | When providing users with fewer location-sharing options there was an increase in the number of users choosing the option(s) that are subjectively closest to the removed option. |

*(Continued)*

**Table 3.** (*Continued*)

| Author | Intervention | Main Findings |
|---|---|---|
| Krol, K., & Preibusch, S (2016) | Warning dialogues | Warnings mentioning security or privacy threats both significantly reduced the disclosure of personal information in the web forms. |
| Kroschke, M., & Steiner, M (2017) | Reviews, peers' behavior | Reviews and peers' behavior both influence information disclosure intention, with the latter having a stronger influence. |
| Larose, R., & Rifon, N. (2007) | Warning labels and privacy seals | Warnings decreased disclosures while seals increased disclosure intentions. |
| Lee, D., Larose, R. (2011) | Personalized social cues: immediacy in the website | Participants' exposure to the high-immediacy level on the site increased their information disclosure intentions |
| Meier, Y., Schäwel, J., Kyewski, E. and Krämer, N. C. (2020). | Fear appeals (warning), social norms | Neither fear appeals or social norms resulted in enhanced privacy protection behavior. |
| Monteleone, S., Bavel, R. Van, Rodríguez-Priego, N., & Esposito, G (2015) | Visceral notices | Anthropomorphic images can increase subjects' predisposition to disclose personal information. |
| Mothersbaugh, D. L., Foxx, W. K., Beatty, S. E., & Wang, S (2012) | Perceived customization benefits, level of information control | Information control and perceived customization benefits both positively influence willingness to disclose personal information. |
| Mukherjee, S., Manjaly, J. A., & Nargundkar, M (2013) | Monetary cues | Priming money increases both the reported willingness and the actual disclosure of personal information. |
| Nosko, A., Wood, E., Kenney, M., Archer, K., De Pasquale, D., Molema, S., & Zivcakova, L. (2012) | Priming story | Participants reading certain priming stories may be encouraged to alter the way in which they display or share personal information. |
| Peer, E., & Acquisti, A (2016) | Reversibility cue | When reversibility is made salient beforehand, people seem to treat the questions as more sensitive and disclose personal information more carefully, consequently providing less disclosing responses. |
| Rodríguez-Priego, N., & Van Bavel, R (2016) | Design of security messages | Long security messages and message accompanied by a male anthropomorphic character led consumers to disclose less personal information |
| Rodríguez-Priego, N., van Bavel, R., & Monteleone, S. (2016) | Design of search engine | The nudges did not lead to differences in the amount of personal information disclosed. |
| Rudnicka, A., Cox, A. L. and Gould, S. J. J. (2019) ' | Motivational message | Motivational messages can increase individual willingness to share personal information |
| Sah, Y. J., & Peng, W (2015) | Visual and linguistic anthropomorphic cues | The direct effect of visual cues was insignificant; yet, there was an indirect effect on information disclosure. Linguistic anthropomorphic cues had positive effects on social perception and information disclosure. |
| Samat, S., Acquisti, A., Clara, S., & Acquisti, A. (2017) | Privacy notices | Participants are significantly less likely to share their personal information when the privacy notice is presented under a 'Prohibit [disclosure]' frame, as compared to an 'Allow [disclosure]' frame. |
| Spottswood, E. L., & Hancock, J. T. (2017) | Visual cues | Explicit cues and surveillance primes can affect disclosure frequency on an SNS. |
| | | The effects of the surveillance primes were subtler, but when they were present they increased disclosure frequency overall. |
| Sundar, S (2013) | Benefit and fuzzy boundary heuristic priming, personalization cues | Individuals who were primed with the fuzzy boundary heuristic were less likely to disclose their information than other conditions. |
| Vitale, J., Tonkin, M.,Ojha, S., Williams, M.-A. (2018) | Embodied robot or disembodied kiosk, transparency | Comparing the transparent and not transparent interfaces within the same system (i.e. robot or kiosk), there are no significant differences in the amount of private information collected by that system. |
| Wang, J., Wang, N., & Jin, H (2016) | Data obfuscation options | Users are more likely to release data when the obfuscation option is available, except for locations data. |
| Wang, N., Zhang, B., Liu, B., & Jin, H (2015) | Privacy notice dialogs | Ads awareness significantly affects actual disclosure behavior. |
| Wang, Y., Leon, P. G., Acquisti, A., Cranor, L. F., Forget, A., & Sadeh, N. (2014) | Visual cues and time delays | Reminders about the audience of posts can prevent unintended disclosures. |
| Zhang, B., & Xu, H (2016) | Frequency and social nudges | Participants felt significantly more comfortable to let the app use their data when they saw the social nudge than the frequency nudge. |

(*Continued*)

**Table 3.** (Continued)

| Author | Intervention | Main Findings |
|---|---|---|
| Zhu, F., Carpenter, S., & Kulkarni, A (2012) | Rational Exposure model—interface | Rational Exposure model did help the participants expose less identity information. |
| *Information (Education)* | | |
| Aiken, K. D., & Boush, D. M (2006) | Trust signals (trustmarks) | A trustmark influences trust that influences a person's willingness to provide personal information while third party certification is the most effective method for developing trust. |
| Feri, F., Giannetti, C., & Jentzsch, N (2016) | Breach notifications | Notifications induce individuals to disclose less information to a firm (those with personally sensitive information). |
| Junger, M., Montoya, L., & Overink, F. J. (2017) | Priming and warning leaflet | Priming and warnings did not prevent disclosure. |
| Mamonov, S., & Benbunan-Fich, R (2018) | Information security threats (news stories) | Exposure to information security threats has positive effect on refusal to disclose sensitive information. |
| Marreiros, H., Tonin, M., Vlassopoulos, M., & Schraefel, M. C (2017) | Privacy messages | Whenever information is about privacy, the type of information (positive or negative) does not matter, while information not mentioning privacy increases disclosure of personal data. |
| Molina, M. D., Shyam Sundar, S. and Gambino, A. (2019) | VPN symbol, Terms and conditions | The provision of a VPN symbol promotes information disclosure, while Terms and Conditions inhibits data sharing. |
| Smith, K. H., Méndez Mediavilla, F. A., & White, G. L (2018) | Facebook privacy training | Participants taking part in a Facebook training shared less personal information. |
| Tsai, J., Kelley, P., Drielsma, P., Cranor, L., Hong, J., & Sadeh, N (2009) | Feedback | People who receive feedback become more comfortable with sharing their location information. |
| Zhang, B., Wu, M., Kang, H., Go, E., & Sundar, S. S (2014) | Security warnings and instant gratification cues | Security cues affect disclosure intention; adding a security cue could trigger more disclosure of social media information while instant gratification cues have no effect on disclosure. |
| *Combination* | | |
| Craciun, G. (2018) | Choice defaults, social consensus | Hearing about peers' behavior, individuals are more likely to share their personal information. Also, respondents were less likely to share in the opt-out default condition. |
| Frey, R. M., Bühler, P., Gerdes, A., Hardjono, T., Fuchs, K. L., & Ilic, A (2018) | Standard privacy policy, customer empowerment, blockchain supported system, monetization | Participants shared similar amounts of personal data for blockchain-supported approaches and standard privacy policies. |
| Gabisch, J. A., & Milne, G. R. (2013) | Safety cues and rewards | Safety cues are more effective than rewards in encouraging information disclosure. |
| Huang, N., Hong, Y., Chen, P.-Y., & Wu, S.-Y. (2018) | Nudging messages: simple request, monetary incentive, relational capital and cognitive capital (framing) | Nudging messages with monetary incentives, relational and cognitive capital framings lead to increase in social sharing behavior, while nudging messages with simple requests decreased social sharing. |
| Hui, K., Teo, H., & Lee, S (2007) | Privacy assurance (privacy statements, and privacy seals), monetary incentives, and information request | The existence of a privacy statement induced more people to disclose their personal information to a website. Monetary incentives have a positive influence on disclosure. Amount of information requested has a negative influence on disclosure. |
| Hutton, L., Henderson, T., & Kapadia, A. (2014) | Monetary incentives, feedback | People are comfortable with disclosing their location for a cash incentive. Participants who received more feedback were much more comfortable with the disclosure of their personal information. |
| Knijnenburg, B. P., & Kobsa, A (2013) | Type of disclosure justification messages, order of requests | Justification messages do not increase disclosure. Changing the request order increases the disclosure of the data requested first but decreases disclosure of data requested later in the interaction. |
| Knijnenburg, B., & Kobsa, A (2016) | Granularity of categories, presentation order, defaults, exceptions | Defaults and order affect sharing of personal information while granularity has no effect on sharing. |
| Mettler, T., & Winter, R. (2016) | Social design features, incentives | Applying social features in ES is highly context dependent. Users are more willing to share when they are promised some type of reward. |
| Preibusch, S., Krol, K., & Beresford, A. R (2013) | Mandatory fields, compensation | Making some fields mandatory jeopardized voluntary disclosure for the remaining optional fields. Monetary incentives for disclosing those same fields yielded increasing revelation ratios for other optional fields. |

*(Continued)*

**Table 3.** (Continued)

| Author | Intervention | Main Findings |
|--------|--------------|---------------|
| Premazzi, K., Castaldo, S., Grosso, M., Raman, P., Brudvig, S., & Hofacker, C. F. (2010) | Compensation of different types, trust (excerpt) | Participants did not claim to be more willing to provide information in the presence of incentives, but in fact, as indicated by their behavior, were more inclined to do so. |
| Warberg, L., Acquisti, A. and Sicker, D. (2019). | Opt-in, opt-out, social norms, message framing | Effects for tailored privacy nudges are difficult to identify. |
| Weydert, V., Desmet, P. and Lancelot-Miltgen, C. (2020) | Monetary compensation, control over data | Offering control of data can increase one's willingness to share personal information while monetary compensation negatively affects data sharing. |
| Xie, E., Teo, H. H., & Wan, W (2006) | Privacy notices, rewards, reputation | All nudges greatly influenced consumers' intention to provide accurate personal information over the Internet, and such effects vary according to the sensitivity of the requested information. |
| *Defaults* | | |
| Baek, Y. M., Bae, Y., Jeong, I., Kim, E., & Rhee, J. W (2014) | Framing of consent forms (opt in/opt out) | The opt-in frame is better at protecting people's information privacy than the opt-out frame |
| Knijnenburg, B. P., Kobsa, A., & Jin, H (2013) | Auto completion tools | The alternative auto-completion tools make people more considerate of the website's purpose in their disclosure decisions. |
| Lai, Y.-L., & Hui, K.-L (2006) | Choice frame and defaults (opt in, opt out) | The "choice-frame, unchecked-default" combination may escalate the level of participation as compared to the "rejection-frame, checked-default" combination in the opt-in context. |
| Tschersich, M (2015) | Default privacy settings | Restrictive default privacy settings lead users in are sharing less personal information to larger group of people. |
| *Incentives* | | |
| Halevi, T., Kuppusamy, T. K., Caiazzo, M., & Memon, N (2015) | Financial incentive | Most participants were not willing to share their fingerprints with an e-commerce application for any feasible reward. |
| Li, H., Sarathy, R., & Xu, H (2010) | Monetary rewards | Monetary rewards may undermine information disclosure intention. |
| Lu, Y., Ou, C. X. J., & Angelopoulos, S (2018) | Monetary incentives or simple reminder | Monetary incentives work no better than reminders in motivating users to disclose personal information. |
| Steinfeld, N (2015) | Monetary rewards | A strong significant correlation was found between the sum of money offered and participants' willingness to grant access to their Facebook profile. |

Several papers found that the use of warnings can reduce actual information disclosure of users [6, 20–25]. Similarly, several studies found that priming, either in the form of privacy-related articles, stories, or videos, was effective at encouraging individuals to limit the sharing of personal information [26–28]. Moreover, a number of papers have investigated the effects of visual cues for privacy decision making; results vary depending on the cue being used by each paper, such as anthropomorphic [29] or monetary images [30]. The majority of papers agree that visual cues can increase the propensity of an individual to disclose personal information [29–32]. However, one research paper argues that the use of visual cues in the form of reminders about the audience of posts can reduce unintended user disclosures [33].

There is a considerable interest in the effects of different design features on individuals' willingness to disclose personal information. A number of papers have empirically demonstrated that certain design features, such as variety of customization options [34], high immediacy levels [35], perception of information control [34, 36], exposure to calls of action [37], videos including female narrators combined with vibrant colors and light musical tones [38], a dynamic privacy score [39], norm-shaping design patterns [40], as well as contextual cues amplifying or downplaying privacy concerns [41], can increase the disclosure of personal information. Also, studies have shown that persuasive messages that are more positively framed or include higher argument strength [42] as well as motivational messages can increase the disclosure of personal sensitive information [43]; while initially participants reacted

negatively to the use of hyper personalization in advertisements, this did not discourage them from sharing sensitive personal information [44].

Meanwhile, other research has failed to show that presentation nudges, including changes in the design of search engines [45], hard to read fonts [46] and increased transparency in the provision of information regarding data processing [47], have an effect on the amount of information being shared during privacy decision making. Contradictory findings are offered by other studies investigating the effects of design features, such as the use of an anthropomorphic character and the length of security messages embedded in notification messages, showing that they can effectively reduce the amount of personal sensitive information that an online consumer is sharing [48].

Moreover, there has been a growing interest in investigating the effect of social nudges, which refer to social framing and peers' behavior pressure aimed to influence users' privacy decision making. Research findings agree that social nudges, revealing that the majority of users' peers have engaged in a similar behavior such as divulgence of personal information [49, 50], usage of applications [51], or adjustment of privacy settings [52], can increase information disclosure. Furthermore, research has shown that adding an obfuscation option, allowing users to share sensitive data such as physical activity and GPS location in anonymized form, can increase the sharing of personal information [53, 54], while a reversibility option, being able to go back and delete previously entered information before submitting, can reduce individual information disclosure [55]. The use of other social nudges, such as presenting profile images of the audience of a social network or numerical information about the audience of a social network, was shown to positively affect people's posting behavior resulting in users making 'better' choices by posting more messages on a private network rather than in a public one [56]. While information about other users (social norms) displayed by a privacy tool did not enhance the privacy protection behavior of Facebook users [57].

In addition, there has been a surge of interest in investigating the role of privacy policies and notices as effective aids on privacy and disclosure decisions; studies have demonstrated that increases in the permissiveness [58] as well as vagueness [59] of a privacy policy can reduce the willingness of an individual to share his/her personal information, while simplification techniques failed to impact disclosure [7]. Moreover, the impact of different degrees of data protection on intention and actual information disclosure has been examined, with results revealing that both normative and objective factors (objective and relative changes in privacy protection) in privacy notices can influence information disclosure [60]. However, in a series of experiments, Adjerid et al. [61] demonstrated that the impact of privacy notices on information disclosure is variable; in their first study results showed that privacy notices implying lower protection actually decreased disclosure, thus achieving the desired support and protection of users, while in the second study findings revealed that the effect of a riskier privacy notice can be significantly reduced by minimal distractions such as a 15 second delay between the notice and the disclosure decision.

Research has also investigated how fine grained sharing options can affect users' disclosure decisions, revealing that when finer options are removed users tend to choose the closest option thus increasing their disclosure rather than erring on the safe side and choosing not to share at all [62]. Meanwhile, the display of finer grained information embedded in data requests lowered users' likelihood to share personal information [63]. Aiming to compare the effects of different interfaces, Vitale et al. [64] found no significant differences in the amount of personal information disclosed when a kiosk or a robot were employed, while Zhu, Carpenter and Kulkarni [65] observed that an interface providing privacy suggestions to users helped them in limiting unnecessary disclosures.

**3.3.2. Nudging with information.** Nudging with information aims at educating and creating awareness in users about the risks and benefits associated with privacy and security decisions using effective communication messages. Provision of information includes two approaches: education and feedback. Education includes the provision of information to users before their engagement with the system thus referring to future decisions, while feedback refers to the information provided alongside the use of the system. Nine papers were identified and included in the present systematic review using the provision of information aiming at assisting and supporting privacy and security decision making. Overall, eight papers have used nudging with information methods and only one study has used feedback. Examples of education nudges include privacy policy documents, notifications, and privacy notices. The papers included have emerged during the years 2006–2020, with most of them being published after 2014, thus demonstrating that only recently has there been an increased interest in information interventions to influence disclosure.

Results of the included papers seem to be inconsistent regarding information priming towards changing privacy and security decisions. A number of papers found no evidence that priming nudges such as warnings and data notifications are effective at reducing information disclosure of personal information [66–68]. However, recent research found that privacy messages prompting users to think about privacy and security issues induced them to share less personal information [69, 70] while the provision of terms and conditions inhibited information disclosure [71]. In the same vein, research findings indicate that users participating in Facebook training shared less personal information [72], while trustmarks and third party certification are the most effective tools towards gaining users' trust and consequently affecting their sharing behavior [73]. Regarding the provision of feedback as a nudging intervention, results indicated that users who received feedback were more comfortable towards sharing their location information [74].

**3.3.3. Nudging with defaults.** Nudging with defaults is defined as intervention aiming to influence users' privacy decisions by setting default options that best serve and align with users' privacy needs and expectations [1]. Four papers were identified in the present systematic review employing the use of defaults as nudging strategy towards influencing privacy decisions. The papers were published between 2006–2015, demonstrating that during the last decade the use of defaults has received attention and becoming an area of interest in the privacy decision making field. Results of the papers show that the use of traditional auto completion tools, usually found in web forms, could cause significantly more information being disclosed [75], while restrictive default privacy settings helped users to share less information on online social networks [76]. Also, the use of an opt-in frame, comparing to an opt-out frame, for the provision of consent proved to be a more effective strategy towards protecting users' privacy [77] while pre-selected options within a choice frame may increase user participation in online activities such as sign up [78].

**3.3.4. Nudging with incentives.** Nudging with incentives aims at motivating users to behave according to their stated preferences. This nudging intervention can take the form of either rewards or punishments. Four papers using incentives as the main nudging strategy were included in the final selection of the present systematic review. The publication years span from 2010 until 2018; all four papers focus on investigating the effects of monetary incentives on privacy decisions in general information disclosure contexts [79, 80] as well as in more specific contexts such as sharing biometric data [8] and granting Facebook access [81]. Research has shown that the higher the amount of the monetary incentive to encourage disclosure, the higher the percentage of people who shared their information [81]. However, other authors argue that, in contrast with the conventional view, most people are not open to sharing their personal data for any feasible reward [8]. Consistent with this view, some studies found

thatfinancial rewards were not effective at motivating users' disclosure [80] or that monetary rewards can actually undermine information dislcosure intentions [79].

It is noteworthy that there is a widespread confusion and considerable debate regarding the relation between nudges and incentives and how nudges can be properly distinguished from other interventions [82]. Thaler and Sunstein (2008) defined a nudge as "any aspect of the choice architecture that alters people's behavior in a predictable way without forbidding any options or significantly changing their economic incentives" [3](p.6). Thus, following this definition, incentives may not be classified as nudges. However, Thaler and Sunstein also introduced the acronym NUDGES as an easy reminder of the six principles for good choice architecture, with the first principle referring to incentives. According to Thaler & Sunstein (2008), the six principles for good choice architecture can be viewed as the acronym NUDGES: iNcentives, Understand Mappings, Defaults, Give Feedback, Expect error, and Structure complex choices. Saghai (2013) amended the definition of nudges by clarifying what it meant by preservation of freedom of choice and by elaborating the importance of substantial noncontrol in nudges: "A nudges B when A makes it more likely that B will φ, primarily by triggering B's shallow cognitive process, while A's influence preserves B's choice-set and is substantially noncontrolling (i.e., preserves B's freedom of choice)" [83](p. 491). Saghai (2013) further suggested that while nudges are substantially noncontrolling, incentives can be substantially controlling or substantially noncontrolling, depending on the valuation (magnitude) of its benefits.

As previously mentioned, the present work follows the classification of nudging for privacy behavior grounded on the seminal work of Acquisti et al. (2017) [1]. In their work, Acquisti et al. (2017) [1] argued that rewards (and punishments) can be used as tools to overcome cognitive and behavioral biases, such as hyperbolic discounting, that affect privacy decision making. All four studies included in the present review offered monetary incentives to participants, either as discount coupon or compensation, ranging from $0.20 to $50, in exchange for users' personal information. Their results indicate that incentives can both increase and decrease disclosure (or have no effect). Following Saghai's (2013) [83], the incentives used in these studies may be considered nudges as their effect on disclosure behavior is not explained by the magnitude of the incentives. Overall, the present work does not seek resolve theoretical debates centered on how to distinguish different (nudging or non-nudging) interventions. Rather, we take a pragmatic approach drawing on an existing classification framework [1] to provide a comprehensive overview of the literature, leaving it for future research to revisit and potentially refine the classification of nudges for privacy decision making.

**3.3.5. Combination.**   A number of papers have used nudges belonging to more than one of the previously mentioned categories (i.e., presentation, information, incentives, and defaults). These are presented below.

There is a small but growing body of academic literature focusing on comparing the effects of rewards, mainly monetary incentives, with presentation nudges, mostly including design features. Most research agrees that the combination of design features with monetary incentives constitute an effective strategy towards enhancing information disclosure. More specifically, papers have demonstrated that embedding social design features within enterprise social systems combined with the provision of monetary incentives positively influenced users' attitudes to share personal information [84], while the use of nudging messages combined with monetary incentives resulted in increased social sharing behavior [85]. Aiming to evaluate the role of privacy assurances as well as the risk-benefit trade off in the context of consumer information disclosure, findings have shown that privacy statements/notices along with monetary rewards can motivate people to share more personal information [86] as well as provide accurate identifiable information [87]. More recently, emerging literature demonstrates that the

use of rewards and safety cues has varying effects on privacy decision making as privacy assurances, presented with safety cues, are more effective over monetary rewards in encouraging information disclosure [88]. Moreover, it has been empirically demonstrated that mandatory fields within a web form could decrease voluntary sharing of personal information for the remaining optional fields, while the provision of monetary incentives for the same fields could increase the rates of disclosure [89].

Research has investigated the effects of information and incentive nudges aiming to understand their impact on disclosure outcomes. Results have shown that among different groups of nudges, including standard privacy policies, customer empowerment, blockchain supported privacy policy and monetization, participants shared similar amounts of personal information [90]. Further it has been demonstrated that a mix of social and financial incentives, such as feedback and cash incentives, were effective strategies in motivating users towards information disclosure [91]. Moreover, investigating the combined effects of trust and compensation in the form of an excerpt describing the e-commerce company and several types of benefits, respectively, Premazzi *et al.* [92] revealed that in low privacy conditions incentives improve information disclosure, while in high privacy conditions compensation decreases actual sharing of information. Control over one's data make people more comfortable with data sharing, while monetary compensation actually decreased people's willingness to share their data with an unknown third party data broker [93].

There has been increased interest in the experimental investigation of the role of default and presentation nudges aiming to influence privacy decision making and sharing of personal information. It has been empirically demonstrated that one's profile information in an social network that is set up under a shared-by-default setting, comparing to a private-by-default setting, could increase sharing of personal information while the granularity of categories had no effect on users' sharing tendency [94]. Moreover, evidence has been observed that the opt-in default settings as well as social consensus constitute powerful mechanisms in increasing information sharing [95]. While in their study examining the potential impact of privacy nudges being tailored to users' personality traits, Warberg, Acquisti, and Sicker [96] used a variety of nudges such as opt-in/opt-out, framing and social norms, demonstrating that it is very difficult to tailor nudges to users' characteristics.

At last, the present review identified only one study focusing on investigating the impact of information and presentation nudges together on privacy decision making. It has been shown that justification messages could lower users' disclosure rates, while the order of data requests could increase the disclosure of data that is requested first and decrease the disclosure of data that is requested later [97].

### 3.4. Meta-analysis

A meta-analysis was conducted using RevMan 5.3 [98] to quantify the effects of nudges on information disclosure decisions. The meta-analysis aims to combine all effect sizes derived from individual studies, resulting in an estimate of the overall effect size regarding the outcome in question (i.e., information disclosure). All 78 papers were initially considered for a meta-analysis. However, 24 papers failed to report sufficient statistical information required for the computation of effect sizes and thus were excluded [99]. As a result, 54 papers were included in the meta-analysis. Nine papers reported more than one experimental study, and 23 studies employed more than one intervention, resulting in a total of 68 studies with 118 effects. Table 4 presents the specific nudges used in the included studies (total number of nudges = 84). All classifications and coding of effect sizes were performed by one co-author and two independent research assistants. Disagreements were resolved either through discussion or by

**Table 4. Nudges in 54 papers included in the meta-analysis.**

| Nudge | Total |
|---|:---:|
| Cues | 13 |
| Warnings | 10 |
| Messages | 10 |
| Privacy notice | 8 |
| Privacy policy | 7 |
| Peers | 6 |
| Design | 5 |
| Requests | 4 |
| Default | 4 |
| Feedback | 3 |
| Order | 3 |
| Seals | 3 |
| Settings | 2 |
| Interface | 2 |
| Benefits | 2 |
| Information control | 1 |
| Training | 1 |

consulting a third coder. Please note that we denote with *n* the number of papers, *i* the number of studies, and *k* the number of effect sizes.

Effect sizes were expressed as standardized mean differences (Hedges' adjusted *g*), which can be interpreted similar to Cohen's *d*, but include an adjustment for small sample bias. Since the true effect size in the population stands to differ between studies employing different interventions, we opted for a random effects model using an inverse-variance method to combine results from different studies. We classified interventions as those seeking to decrease disclosure (57.3%) and those seeking to increase disclosure (40.8%), based on predictions put forward by the original author(s) (one study encompassing two interventions did not report any predictions). To enable a synthesis of all interventions, positive effect sizes denote predicted differences between a treatment and control group (i.e., an increase in disclosure when an increase was predicted, or a reduction in disclosure when a reduction was predicted), and negative effect sizes denote unpredicted differences (i.e., an increase in disclosure when a reduction was predicted, or a reduction in disclosure when an increase was predicted). Since the original author(s)'s predictions may not reflect true differences in the population, we supplemented our coding with the *coining* method described by Fanelli, Costas, and Ioannidis [100]. In particular, we re-coded unpredicted effects into predicted effects (i.e., we multiplied effects by -1) when an effect size exceeded a conservative 1.65*SE threshold and thus had a likelihood of occurrence of *p* = .05 (one-sided). This adjustment affected four out of 118 effect sizes (3.4%). In further sensitivity analyses, we repeated all primary analyses using absolute effect sizes (a more liberal approach to *coining*, see ([100]).

As shown in the forest plot depicted in Fig 3, nudging interventions had a small-to-medium sized overall effect on disclosure, Hedges' *g* = 0.32 [0.25, 0.38]. Effect sizes were heterogeneous, $I^2$ = 89%, as anticipated, and re-affirming our decision to employ a random effects model. When looking at different types of nudging interventions, the strongest effect was observed for *incentive* and *default* interventions, Hedges' *g* = 0.42 [0.14, 0.70] and 0.41 [0.24, 0.59], respectively. Meanwhile, the weakest effect was observed for *information* interventions, Hedges' *g* = 0.18 [0.06, 0.31], followed by *presentation* interventions, Hedges' *g* = 0.33 [0.24, 0.42]. Effect

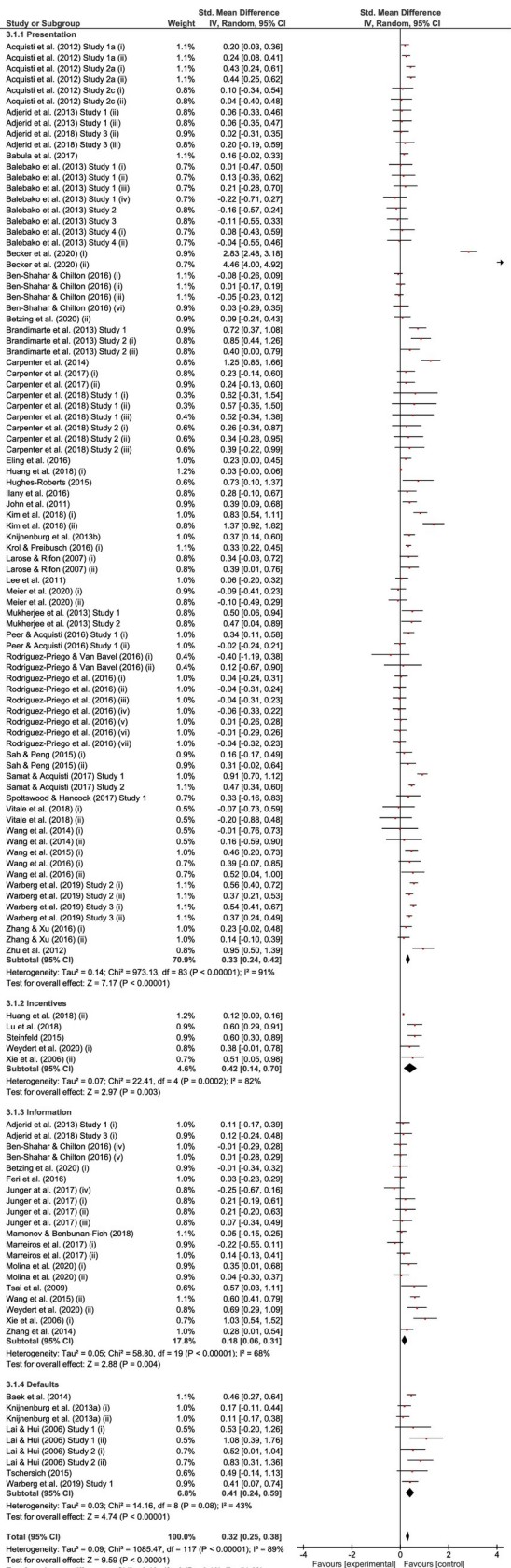

**Fig 3. Forest plot.** *Note*: Where applicable, the number of participants was adjusted by dividing the total number of participants by the number of measures and/or interventions administered to the same group of participants (denoted *(i)* to *(vi)*; see ([10]). This adjustment does not affect the total number of participants. Note that, while study-level confidence intervals are wider when adjustments are made, estimates of central tendency (i.e., standardised mean difference) are unaffected.

sizes do not differ significantly between subgroups, $\chi^2(3) = 6.19$, $p = .10$. Furthermore, pairwise comparisons ($k = 6$) keeping the global error rate at 5% also revealed no significant differences between subgroups. Thus, we refrain from drawing any conclusions regarding the relative strengths of the different interventions (*presentation* vs. *incentive* vs. *information* vs. *default*).

*Secondary analysis.* Overall, interventions to increase disclosure had a stronger impact on disclosure when compared to interventions to decrease disclosure, Hedges' $g = 0.33$ [0.25, 0.40] and 0.20 [0.13, 0.27], respectively; test for subgroup differences: $\chi^2(1) = 6.04$, $p = .01$.

*Sensitivity analysis.* As outlined above, we repeated all analyses, this time employing a more liberal approach to *coining* using absolute effect sizes [100]. This sensitivity analysis yielded the same conclusions, with an overall effect size of Hedges' $g = 0.35$ [0.29, 0.41] across all interventions.

Finally, we also examined funnel plots to gauge the presence of potential reporting bias. As shown in Fig 4 and Fig 5A–5D, there was some evidence for reporting bias for all nudging strategies. However, the number of studies using *incentive*, *information*, and *default* nudges was relatively low, which limits the conclusions that can be drawn from the funnel plots. On the other hand, the number of studies reporting *presentation* nudging interventions was noticeably larger. Here, the funnel plot points to the presence of several studies reporting unexpectedly large effects. Two effects in particular appear to be outliers, with standardized mean differences that exceed the standard error more than fifteen-fold. Effect sizes remain significantly heterogeneous when the two outliers are removed, $I^2 = 81\%$. This is consistent with the observation that the nature of the *presentation* interventions differs widely between studies (see Table 3). Outliers aside, differing interventions may explain the distribution of effect sizes observed for presentation interventions [101]. Crucially, removing large effects relative to the standard error from the meta-analysis (including the two outliers) had a limited impact on the

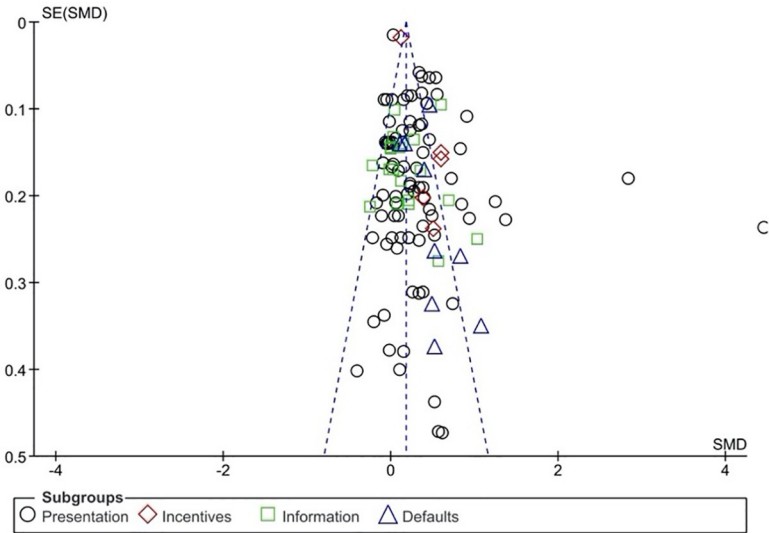

**Fig 4. Funnel plot.**

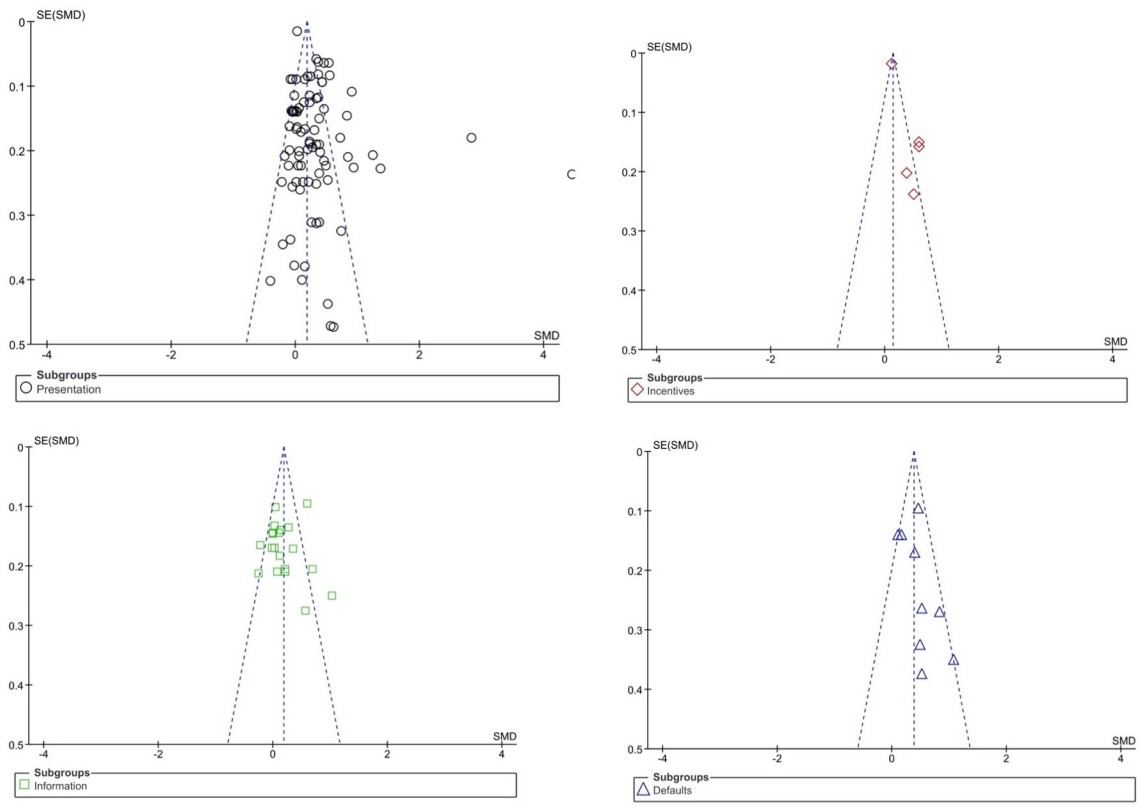

**Fig 5. a.** Funnel plot for presentation nudging. **b.** Funnel plot for incentive nudging. **c.** Funnel plot for information nudging. **d.** Funnel plot for information nudging.

overall effectiveness of presentation interventions, which remained significant. Similarly, removing outliers did not impact the conclusions derived from the primary and secondary analyses.

*Discussion*. There is a growing body of research investigating the impact of presentation nudges on users' information disclosure. In our meta-analytic review, the majority of effects (84 out of 118) represent presentation nudging, while the rest of the three categories ($i_{\text{incentives}} = 5$; $i_{\text{information}} = 20$; $i_{\text{defaults}} = 9$) account for less than 30% of the total effect sizes. Presentation nudges ranged from warnings and design features, to social nudges and visual cues. Even within those sub-types, presentation nudges vary. For example, studies examining design features have experimented with information layout, different fonts, anthropomorphic characters, and so forth. It might be that different sub-categories of presentation nudges show differential effects on information disclosure. Future research studies should aim at exploring in more depth the impact of different presentation nudges. For example, in the case of visual anthropomorphic cues, studies may consider race and gender as well as different contexts such as a doctor in healthcare, a salesperson in an e-commerce website or a teacher in an online educational platform in order to explain potential differences in findings in existing studies.

Results of the meta-analysis show that in absolute terms incentives had the strongest effect on users' information disclosure.This is consistent with the wider literature on behavioral change. For example, previous meta analytic research indicates that financial incentives are effective in promoting changes in behavior such as encourage pro-environmental behavior (e.g., recycling) [102] and healthy behavior change (e.g., smoking) [103]. However, it should

be noted that the number of studies in this category of interventions was relatively small ($i$ = 5), and as such conclusions should be drawn with caution. A separate meta-analysis excluding papers with incentives has been conducted, which shows no significant changes in the results and hence no changes to the conclusions of this study (see S1 Appendix).

Our findings reveal that although the effects of information nudges were significant, they were (in absolute terms) the smallest out of all nudging strategies. This is consistent with the studies on health behaviors, which found that information campaigns were often either ineffective at changing behaviors [104], or the campaigns backfired promoting undesirable outcomes [105]. According to Thaler (2018) [106], a behavioural intervention that does not encourage a behavior in the individual's best interest is called a 'sludge'. In this review, all selected papers aimed at examining the effect of nudges on disclosure. However, their results sometimes were unexpected.

Furthermore, regarding the effect of default nudges, the results of the meta-analysis revealed a medium-sized effect. These results dovetail recent meta-analytic evidence attesting to the effectiveness of default nudges across a range of outcomes [12]. However, since the number of studies examining information disclosure more specifically remains relatively small ($i$ = 9), some caution is warranted, and further studies are needed to ascertain the effectiveness of default nudges on information disclosure.

Finally, our secondary analysis revealed that interventions aimed at increasing disclosure had a stronger impact on disclosure when compared to interventions aimed at decreasing disclosure. In other words, nudges may be more effective in motivating people to share their personal information rather than discouraging people to withhold such information. This asymmetry may be explained by the presence of a floor effect: if participants are somewhat reluctant at baseline to disclose personal information, then there is more 'headroom' for subsequent interventions to increase (*vs.* reduce) disclosure. Future studies are needed to further explore the potential differential effects of nudging aimed at increasing *vs.* decreasing information disclosure.

## 4. Discussion

### 4.1. Implications

Identifying ways to nudge users into making more informed and desirable privacy decisions is of significant practical and policy value [1]. The main contribution of this systematic meta-analytic review is the comprehensive evidence gathering of the applications and effectiveness of a variety of privacy nudging strategies to influence disclosure of personal information in digital environments. The systematic review of empirical research publications from 2006, which included 69 papers, revealed a growing interest in digital privacy nudges for disclosure of personal information while also unveiling inconsistencies in research findings. In general, three broad conclusions could be derived from the results of the qualitative synthesis and meta-analysis: (1) most empirical papers have focused on nudging with presentation to influence users' privacy decisions, (2) a meta-analysis of 54 papers revealed that nudging strategies can effectively influence information disclosure, (3) further research is needed to elucidate the relative effectiveness of different intervention strategies and how nudges can confound one another.

Firstly, most papers included in this systematic review experimented with presentation nudges to influence disclosure of information. While some of these strategies are closely related to other nudging dimensions (e.g., notices and warnings are related to nudging with information), it is apparent that other nudging dimensions have yet to be tested widely in academic research. The different nudging dimensions are designed to address specific users'

biases and limitations in privacy decision making. Nudging with presentation aims at reducing users' cognitive load, addressing such biases as framing effect, optimism bias, and overconfidence. Some of these biases and limitations can also be addressed by information nudges, to reduce information asymmetry [1]. Other biases, such as status quo bias and habit, are better addressed by default nudges, reducing users' efforts by configuring user interface according to expectation. Therefore, it can be suggested that existing privacy nudging studies thus far have provided more support on ways to reduce users' cognitive load, but less on ways to reduce efforts and increase motivation.

Secondly, evidence on the effects of nudging strategies on disclosure of personal information has been rather mixed and inconsistent in existing literature. However, our meta-analysis of 54 papers testing a variety of nudges demonstrated that overall nudging strategies were effective in influencing disclosure of personal information. These findings align with a recent meta-analysis [12] investigating the effectiveness of a wide range of nudging strategies (e.g., reminders, feedback, simplifications) in various contexts such as health, energy, policymaking, and privacy. However, in contrast to this previous work, we were able to examine the impact of different digital nudging strategies on personal information disclosure, showing that, while all nudging strategies had a statistically significant effect on disclosure, *incentive* and *default* interventions appeared to be particularly effective. However, it should be noted that the number of studies in the latter categories was relatively low. As a result, more primary research on privacy nudges is essential, along with new evidence syntheses to establish the impact of different nudges on information disclosure.

Thirdly, researchers have tested different combinations of multiple nudging strategies simultaneously. This base of research offers important conceptual and practical contributions as the research design allows for the comparison and identification of how one nudge can be used to amplify (or reduce) the effect of another. Practically speaking, using combinations of nudges may offer a "one-two punch" to influence disclosure behavior. From a theoretical point of view, when users are bound by a set of limitations, elucidating the extent to which different limitations (e.g., biases) play a role in privacy decision making (e.g., which will most likely lead to decision heuristics) is important. Research utilizing combination of nudges, designed to leverage or mitigate different users' limitations, will allow for identifying most persistent behavioral problems in privacy decision and thus prioritizing relevant nudging strategies to alleviate them.

This study offers relevant practical and policy implications on a range of intervention strategies that can be used to nudge users into making responsible decisions when sharing personal information. For practitioners wishing to influence disclosure behaviors, this study showed that nudging to increase disclosure may be more effective than nudging to decrease disclosure. In addition, this study revealed which strategies are most likely to be effective in motivating users to share information, such as nudging with incentives (e.g., monetary rewards), defaults (e.g., opting-out option), and (some) presentation options (e.g., calls to action), while highlighting that caution may be advisable with strategies that seek to change behavior by educating users (i.e., information nudges). Importantly, this study contributes to a better understanding of ways to influence information disclosure in digital environments.

## 4.2. Limitations

Despite its contributions, this study has several limitations, which mainly come from variation in research design amongst the studies included in the systematic review. There has been a wide range of interventions strategies used in the studies. While these strategies can be grouped into various categories (i.e., nudging dimensions), some of them are not completely

comparable. For example, different studies experimented with different presentations of privacy warnings, such as warnings with source, framing of warnings, and warning dialogues. The contexts of data disclosure are also varied. Furthermore, studies use different types of outcome variables representing disclosure; some studies used dichotomous variables, continuous variables while others use multiple disclosure outcomes. Also, some of the papers conducting experimental studies have deployed relatively small sample sizes, as a result, generalized conclusions should be drawn with caution. Future research is essential in order to evaluate the effectiveness of a wider range of privacy nudging strategies aiming to identify tried and tested practice for effective privacy nudging.

### 4.3. Future work

Based on the findings and due to the limitations of this systematic review, the following can be suggested to guide the direction of future research in this area. In order to alleviate issues with variations in study designs, future studies should endeavor to experiment using the same intervention approaches in different contexts (e.g., e-commerce vs online health environment), using different interfaces (e.g., embodied vs disembodied agents), and/or with different groups of participants (e.g., older adults). Having more empirical research on specific nudging approaches (e.g., through replication studies) will allow for more rigorous testing of consistency in nudging effects on disclosure. This can also be achieved by ensuring consistent use of outcome variables in experiments with nudging by encouraging the use of standardized, validated measures of disclosure behavior. Furthermore, it is important for future research to focus on conducting experiments testing actual information disclosure, instead of behavioral intentions, in real settings in order to provide more robust and generalizable results. Most importantly, the results of this systematic review indicate that the quest for best practices in privacy nudging should continue. For example, empirical studies experimenting with defaults as nudging strategies are relatively scant as most research has focused on presentation nudges. Also, reversibility and timing nudges have been largely neglected from existing empirical research. Overall, it is important for future research to experiment with a richer variety of interventions in order to inform best practices of nudging for privacy. Lastly, future research should focus on the refinement of the privacy nudging framework provided by Acquisti et al. [1]. Our review has uncovered significant variation in the category of presentation nudging. Within-category differences make it difficult to understand and identify the potential varying impact of different presentation nudges (e.g., visual, social, and linguistic cues) on information disclosure.

### 4.4. Conclusion

Digital nudging has been mooted as a tool to alter user privacy behavior. The present work offers comprehensive evidence on the influence of various nudging strategies on the disclosure of personal information in digital environments. The systematic review of 78 papers showed that the majority of empirical research has focused on presentation nudges to influence users' privacy decisions; while the meta-analysis of 54 papers revealed that interventions aiming to increase disclosure is more effective than those aiming to decrease disclosure. Further research is needed to continue the quest for best practices in nudging.

## Supporting information

**S1 Checklist. PRISMA 2009 checklist.**
(DOC)

**S1 Appendix.**
(DOCX)

**S1 Protocol.**
(PDF)

## Author Contributions

**Conceptualization:** Athina Ioannou, Iis Tussyadiah, Mario Weick.

**Formal analysis:** Athina Ioannou, Iis Tussyadiah, Shujun Li, Mario Weick.

**Funding acquisition:** Iis Tussyadiah, Graham Miller, Shujun Li, Mario Weick.

**Investigation:** Athina Ioannou, Iis Tussyadiah.

**Methodology:** Athina Ioannou, Iis Tussyadiah, Mario Weick.

**Project administration:** Iis Tussyadiah, Graham Miller, Shujun Li.

**Supervision:** Iis Tussyadiah, Graham Miller, Shujun Li.

**Validation:** Iis Tussyadiah.

**Visualization:** Athina Ioannou, Mario Weick.

**Writing – original draft:** Athina Ioannou, Iis Tussyadiah, Mario Weick.

**Writing – review & editing:** Athina Ioannou, Iis Tussyadiah, Graham Miller, Shujun Li, Mario Weick.

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
