## [Decision Letter · Decision Letter 0]

13 Apr 2021

PONE-D-21-06378

Privacy Nudges for Disclosure of Personal Information: A Systematic Literature Review and Meta-Analysis

PLOS ONE

Dear Dr. Ioannou,

Thank you for submitting your manuscript to PLOS ONE. The paper has been carefully considered and appreciated, but does not fully meet PLOS ONE’s publication criteria as it currently stands. Therefore, we invite you to submit a revised version of the manuscript that addresses the points raised during the review process.

Particular care during the revision should be devoted to thoroughly address reviewers’ methodological comments. Substantial improvements are expected and required for a positive evaluation. For each of the points raised by the reviewers, a specific comment is required.

Specifically, Reviewer 1 raises a major concern with regard to the choice of almost completely rely on  Acquisti et al.’s (2017) taxonomy of nudges. Reviewer 2 comments on some limitations of that taxonomy and suggests extensions. This seems a particularly relevant point calling for a detailed examination.

Reviewer 2 presented several observations that require careful consideration in order to improve the paper.

Both reviewers commented on several imprecisions in quoting and in citations that should be fixed in the next version.

We look forward to receiving your revised manuscript.

Kind regards,

Marco Cremonini, Ph.D.

Academic Editor

PLOS ONE

Journal Requirements:

2. Please include captions for your Supporting Information files at the end of your manuscript, and update any in-text citations to match accordingly. Please see our Supporting Information guidelines for more information: http://journals.plos.org/plosone/s/supporting-information

Reviewers' comments:

Reviewer's Responses to Questions

**Comments to the Author**

1. Is the manuscript technically sound, and do the data support the conclusions?

Reviewer #1: Yes

Reviewer #2: Yes

2. Has the statistical analysis been performed appropriately and rigorously? 

Reviewer #1: I Don't Know

Reviewer #2: Yes

3. Have the authors made all data underlying the findings in their manuscript fully available?

Reviewer #1: Yes

Reviewer #2: No

4. Is the manuscript presented in an intelligible fashion and written in standard English?

Reviewer #1: Yes

Reviewer #2: Yes

5. Review Comments to the Author

Reviewer #1: This paper presents a meta-analysis of literature on digital nudging. It is well written, coherent, clear, and well structured. It steers clear of the question regarding the overall effectiveness of digital nudging, as this would require knowing about the studies with a null result that were never published. In other words, the paper shows an awareness of publication bias in experimental studies. The paper therefore limits itself to an analysis the corpus of studies that it collected using clear and consistent criteria. I enjoyed reading it and I believe it can make an important contribution to the literature on digital nudging.

General comments

I have one major concern. The authors rely on Acquisti et al.’s (2017) taxonomy of nudges, and group their studies according to the categories of information, presentation, defaults, and incentives. They neglect reversibility (error resiliency) and timing, which they should acknowledge and possibly explain. Peer and Acquisti, 2016, for example, included in the meta-analysis, would seem to fit under reversibility. However, the main problem lies with the category ‘incentives’.

The whole point of nudging is to go beyond ‘traditional’ policy approaches to influencing behaviour, like bans or taxes. To influence behaviour by offering an incentive is not behavioural economics at all, it is simply economics. It is the most basic, traditional economic approach to influencing behaviour. This is why the demand curve slopes downward and the supply curve slopes upward: because price matters. A monetary incentive, therefore, should not be considered a nudge. The authors acknowledge this when they quote Thaler and Sunstein’s (2008) definition of a nudge on page 3: a nudge is ‘any aspect of the choice architecture that alters people’s behavior in a predictable way without forbidding any options or significantly changing their economic incentives.’

I realise the authors are relying on an existing taxonomy, namely Acquisti et al.’s. But it seems to me, from a cursory glance, that Acquisti et al. had a more nuanced view of incentives. In some cases, rewards were non-monetary. In others, they intended to offset existing biases, such as hyperbolic discounting. Still, in other cases, biases such as loss aversion were leveraged to shape the perception of rewards and punishments. I would invite the authors to reflect on this problem. If the use of the incentive category is justified (à la Acquisti et al.), they should explain this. If it is not, the incentive category and the four papers within it should be excluded.

Specific comments

- The paper is a bit sloppy when quoting references. For one, it seems to have been written with a different referencing system in mind (Harvard or APA), which sometimes leads to inelegant phrasing, e.g. ‘[8] sought to nudge users…’ Acquisti et al. (2017) appears in the text without its reference number. On pages 21 and 22, references 38 and 65 are mentioned twice. The paper would benefit from a proofreading throughout to address these issues.

- On page 11, the paper claims ‘the investigation of privacy nudges has grown significantly during the last decade, since 2013’. Strictly speaking, given the time lag in scientific publications, the growth in investigation probably began in 2011 or 2012.

- In Section 3.3.4, first par: ‘Research has shown that the higher the amount of the monetary incentive, the higher the percentage of people who shared their information [81].’ Monetary incentive for what? For disclosing or not disclosing? Common sense would dictate that it is for disclosing, but this should be clarified.

- On page 31, the paper claims ‘results of the meta-analysis show that in absolute terms incentives had the strongest effect on users’ information disclosure’, which is not surprising, as incentives work on a different level (as I noted above). I wonder if this strong effect pushes the overall effect of the studies over a certain threshold. In other words, what would be the overall effect of the studies if we excluded incentives? Given that incentives are not really nudges, it would be good to know this. Would conclusion no (2) ‘a meta-analysis of an adequate papers revealed that nudging strategies can effectively influence information disclosure’ still hold?

- On page 32, the paper claims that the number of studies in the ‘incentive’ category were five, but elsewhere it says they were four. I guess it means four studies, but five effects. Could you please clarify?

Reviewer #2: This is an interesing study. You need to some work on the paper if you want to get it published

(1) People do not make imperfect decisions - to them, the decisions are perfect. They make unwise decisions

(2) You have a habit of not using the author name. eg [1] says .... Respect the author enough to name them

(3) When a nudge encourages unwise disclosure, it is no longer a nudge which must always be for the good of the nudgee. What it becomes is SLUDGE

(4) You are conflating privacy and security. Malware is a security problem. Disclosure is a privacy issue. If malware makes data inaccessible, that is an availabiity issue (from the CIA principles) and thus security.

(5) when quoting, provide page numbers

(6) Personal infomation should include mention of telephone numbers and email addresses

(7) sec 3.4 referencing problem in first line

(8) p26 - last para - what does 84 refer to?

(9) p31 - again, when a nudge is not doing good to the nudgee, it is sludge

(10) provide a reference for incentives. BTW - nudges, per definition, exclude incentives (rewards) or sanctions. Strictly speaking, anything using an incentive is not a nudge.

(11) The conclusions and implications section is FAR too long. Call this reflection and tighten it up. The section rambles too much

(12) add a short and concise conclusion

(13) at the end you say that more work needs to be done. This is standard in future work sections but disatisfactory. Say how - you know the literature so say exactly how this should be achieved.

(14) incomplete refs: 2, 13,, 15, 24, 35, 29, 48, 95 (is xx correct?), 103 - what is CNBC?, 105 (repeats at end), 46 repeats at end

6. PLOS authors have the option to publish the peer review history of their article (what does this mean?). If published, this will include your full peer review and any attached files.

Reviewer #1: No

Reviewer #2: **Yes: **Karen Renaud

---

## [Author Response · Author response to Decision Letter 0]

12 May 2021

We have attached a document with reponses to reviewer comments (see attached).

---

## [Decision Letter · Decision Letter 1]

2 Jul 2021

PONE-D-21-06378R1

Privacy Nudges for Disclosure of Personal Information: A Systematic Literature Review and Meta-Analysis

PLOS ONE

Dear Dr. Ioannou,

Thank you for submitting your manuscript to PLOS ONE. Reviewers have carefully read your manuscript and overall they agree that it clearly improved from the original submission. A remaining open issue, however, still requires further consideration. Therefore, we invite you to submit a revised version of the manuscript that addresses the points raised during the review process.

In particular, Reviewer1 has commented on a clearly relevant issue concerning incentives. The manuscript could benefit from the inclusion of a clear distinction between nudges and incentives, a discussion on the inclusion or exclusion of incentives from models, and in general a clear presentation of the issue and of relevant approaches regarding this key aspect. Reviewer1's suggestion is to give to this discussion more space and more evidence. Authors are invited to further consider this aspect and submit a revised manuscript or provide a detailed rebuttal of the suggestion.

A rebuttal letter that responds to each point raised by the academic editor and reviewer(s). You should upload this letter as a separate file labeled 'Response to Reviewers'.A marked-up copy of your manuscript that highlights changes made to the original version. You should upload this as a separate file labeled 'Revised Manuscript with Track Changes'.An unmarked version of your revised paper without tracked changes. You should upload this as a separate file labeled 'Manuscript'

We look forward to receiving your revised manuscript.

Kind regards,

Marco Cremonini, Ph.D.

Academic Editor

PLOS ONE

Journal Requirements:

Reviewers' comments:

Reviewer's Responses to Questions

**Comments to the Author**

1. If the authors have adequately addressed your comments raised in a previous round of review and you feel that this manuscript is now acceptable for publication, you may indicate that here to bypass the “Comments to the Author” section, enter your conflict of interest statement in the “Confidential to Editor” section, and submit your "Accept" recommendation.

Reviewer #1: (No Response)

Reviewer #2: All comments have been addressed

2. Is the manuscript technically sound, and do the data support the conclusions?

Reviewer #1: Partly

Reviewer #2: Yes

3. Has the statistical analysis been performed appropriately and rigorously? 

Reviewer #1: I Don't Know

Reviewer #2: Yes

4. Have the authors made all data underlying the findings in their manuscript fully available?

Reviewer #1: (No Response)

Reviewer #2: Yes

5. Is the manuscript presented in an intelligible fashion and written in standard English?

Reviewer #1: Yes

Reviewer #2: Yes

6. Review Comments to the Author

Reviewer #1: Thank you for taking my comments into account. Regarding incentives in particular, I am reassured that running the analysis without incentives does not change the main outcome of the paper, which was a big worry. However, I am disappointed that you did not give this issue greater importance in the paper. It is reduced to a footnote and to an analysis in the Supplementary Material. You did not take up my invitation to reflect more on the question of incentives. Just hiding behind Acquisti et al.'s taxonomy as a justification is not enough - in my opinion - to advance scientific knowledge on this issue and move the debate forward. At the very least, you have a glaring contradiction in your paper, which has "Nudges" in its title, cites Thaler and Sunstein's definition that an incentive is *not* a nudge, and then happily embraces Acquisti et al.'s taxonomy without questioning whether the inclusion of incentives is appropriate in this context. I would have expected at least a paragraph where you discuss this issue. You could argue that perhaps it depends on the behaviour being observed. Or you could go into the detail of how Acquisti et al. apply the category of incentives in their study (as I mentioned in my review). Perhaps in the context of privacy an incentive can and should be considered a nudge, in apparent contradiction to Thaler and Sunstein. But the reader would like to know why. I would like to insist on such a reflection on your part. I think it is important, and if we start overlooking these important issues, lying at the core of the behavioural turn in policy-making, the value of the approach will be undermined and it will all start to go downhill from there. Please provide at least a paragraph, possibly in Section 3.3.4, where you address this so that the reader who is interested in nudges and privacy can better understand if - in the context of privacy and following the definition of Thaler and Sunstein - an incentive can or cannot be considered a nudge.

Reviewer #2: I'm happy with the revisions the authors made. They have been responsive to my comments and the paper is much improved

7. PLOS authors have the option to publish the peer review history of their article (what does this mean?). If published, this will include your full peer review and any attached files.

Reviewer #1: No

Reviewer #2: No

---

## [Author Response · Author response to Decision Letter 1]

13 Aug 2021

We thank the Reviewers and Editor for their constructive feedback. We have revised the manuscript addressing the issue at hand. 

We have revised the manuscript to include two paragraphs in section 3.3.4 reflecting on the issue of the inclusion of incentives as nudges in the present paper. We reflect on the disagreement regarding the relation of nudges and incentives in existing literature. We offer justification on potential reasons that incentives can be considered as nudges in the context of privacy. We also highlight this issue as essential future work, and clarify that the present study does not seek to offer theoretical justification on how to distinguish different interventions; but rather aims to offer empirical evidence on the effectiveness of nudging interventions towards influencing privacy decision making.

---

## [Editor Report · Decision Letter 2]

17 Aug 2021

Privacy Nudges for Disclosure of Personal Information: A Systematic Literature Review and Meta-Analysis

PONE-D-21-06378R2

Dear Dr. Ioannou,

The revised paper has successfully addressed all reviewers' comments, therefore we’re pleased to inform you that your manuscript has been judged scientifically suitable for publication and will be formally accepted for publication once it meets all outstanding technical requirements.

Kind regards,

Marco Cremonini, Ph.D.

University of Milan

Academic Editor

PLOS ONE
---

## [Editor Report · Acceptance letter]

19 Aug 2021

PONE-D-21-06378R2 

Privacy Nudges for Disclosure of Personal Information: A Systematic Literature Review and Meta-Analysis

Dear Dr. Ioannou:

I'm pleased to inform you that your manuscript has been deemed suitable for publication in PLOS ONE. Congratulations! Your manuscript is now with our production department. 

Kind regards, 

on behalf of

Dr. Marco Cremonini 

Academic Editor

PLOS ONE